# Serum Immunoglobulin Levels in Children with Sickle Cell Disease: A Large Prospective Study

**DOI:** 10.3390/jcm8101688

**Published:** 2019-10-15

**Authors:** Sophia Cherif-Alami, Isabelle Hau, Cécile Arnaud, Annie Kamdem, Basil Coulon, Elodie Idoux, Stéphane Bechet, Rita Creidy, Françoise Bernaudin, Ralph Epaud, Corinne Pondarré

**Affiliations:** 1Service de Pédiatrie, Centre de Référence de la Drépanocytose, Centre Hospitalier Intercommunal de Créteil, 40 Avenue de Verdun, 94000 Créteil, France; sophia.cherif-alami@chu-dijon.fr (S.C.-A.);; 2ACTIV Association Clinique et Thérapeutique Infantile du val de Marne, 27 rue d’Inkermann, 94100 Saint Maur des Fossés, France; 3Laboratory of Immunology, Centre Hospitalier Intercommunal de Créteil, 40 Avenue de Verdun, 94000 Créteil, France; 4INSERM Unité 955, Paris XII University, 8 rue du Général Sarrail, 94010 Créteil, France

**Keywords:** sickle cell disease, immunoglobulins, spleen

## Abstract

Over the past 3 decades, the pediatric department of the university Intercommunal Créteil hospital, a referral center for sickle cell disease (SCD), has prospectively evaluated immunoglobulin (Ig) levels in a cohort of 888 children with SCD, including 731 with severe sickle genotypes (HbSS and HbSβ^0^ thalassemia) and 157 with milder genotypes (HbSC and HbSβ^+^ thalassemia). We found consistent sickle genotype differences in levels of IgG and IgA, with increased levels of IgA and IgG in the severe versus milder genotype, from early childhood to late adolescence. Additionally, our results revealed a low serum IgM level, irrespective of sickle genotype. Finally, we found that IgA and IgG levels were significantly increased after therapeutic intensification with hydroxyurea but were stabilized in children receiving a transfusion program. The mechanisms contributing to these changes in Ig levels are unclear as is their clinical significance. We believe they should be further investigated.

## 1. Introduction

The spleen is one of the first organs damaged in sickle cell disease (SCD), leading to increased susceptibility to pneumococcal infections and splenic sequestration [1,2]. In the splenic red pulp, red blood cells (RBCs) are not contained in capillaries, so classical vaso-occlusion does not occur. The red pulp is instead congested by RBCs, and congestion is probably caused by markedly impaired deformability and increased adhesion properties of sickle RBCs. These abnormalities play a key role in progressive spleen damage and loss of function [3].

Among sickle genotypes, the severe subgroup (HbSS and HbSβ^0^ thalassemia) is associated with early loss of splenic function. A recent study using 99^m^Technetium (Tc) sulfur-colloid liver–spleen scan to evaluate the phagocytic ability of splenic macrophages in 193 infants aged 8–18 months found that loss of splenic function began before 12 months of age in 86% of infants [4]. Another study, investigating the mechanical filtration ability of the spleen by using 99^m^Tc heated RBC scintigraphy in a cohort of 57 infants with sickle cell anemia, showed that in 32% and 50% of infants, splenic function was decreased at 6 and 18 months of age, respectively [3]. The age varies for irreversible fibrosis of the spleen (a loss also referred to as autosplenectomy), and changes have been described over time, likely due to improved supportive care and the use of acute and chronic transfusion therapy [5]. On ultrasonography, autosplenectomy was found in 39.5% of a cohort of 86 patients with HbSS who were 0–21 years of age but in only 5% of children less than 5 years of age [5].

In milder SCD forms such as HbSC and HbSβ^+^ thalassemia, the degree of splenic impairment is less clear, and few studies have explored spleen function. In 201 patients with HbSC at 6 months to 90 years of age, the number of pitted cells (RBCs also used to measure hyposplenic function) was significantly increased with age. None of the 59 children less than 4 years of age had abnormally high pitted cells, whereas the count increased in 19 of 96 (22%) children with 4–12 years of age and 25 of 56 (46%) older than 12 years of age [6]. In HbSC patients, autosplenectomy was found in only 3 of 21 patients (14.3%) with 0–21 of age [5]. Splenic loss of function occurs with milder SCD genotypes, however later in life, and to a lesser extent than in patients with severe SCD genotypes.

The spleen functions as a phagocytic filter, removing old and damaged cells and blood-borne microorganisms, but it also plays a specific role in immune responses against pathogens, linking innate and adaptive immunity. The spleen is required for the generation and survival of a unique B-cell population residing in the marginal zone (MZ) of the spleen, non-switched immunoglobulin M (IgM) B cells. These MZ B cells provide early T-cell independent responses by producing natural anti-polysaccharide IgMs, which are essential to the phagocytosis of otherwise poorly opsonized encapsulated bacteria [2]. Their number has been shown to be significantly reduced in HbSS disease as compared with patients with HbSC disease or control patients [7]. However, the potential impact on IgM plasma levels in these patients has not been extensively studied.

Finally, the spleen contributes to tolerance to antigens by trapping circulating apoptotic cells. This function is essential to prevent inflammatory pathology and the development of autoimmunity. Of note, antinuclear antibody positivity is more common in SCD, and a higher than expected incidence of connective tissue diseases has been reported in patients with HbSS subtypes as compared with the general population [2,3,4,5,6,7,8,9].

Very few and small studies have explored Ig levels in a SCD population, and strong data are missing. In these earlier studies, IgG and IgA hyperglobulinemia was commonly described, whereas reports of IgM levels were conflicting [10,11,12,13,14]. High IgG and IgA levels may be due to chronic inflammation, but the precise mechanism of the occurrence of IgG and IgA hyperglobulinemia has not been elucidated. Low IgM levels were found correlated with lack of functional splenic tissues in a small cohort of adult patients [10].

In the present work, we sought to investigate IgG, IgA, and IgM profiles by using prospectively collected longitudinal data from the Créteil pediatric SCD cohort.

## 2. Materials and Methods

### 2.1. Study Population

All parents provided informed consent for data to be used for research. Use of the database was approved by the Créteil Institutional Review Board and the French National Data Protection Commission (CNIL, no. 2069568).

All children from the Centre Hospitalier Intercommunal de Créteil (CHIC) cohort were included, with severe sickle genotypes (HbSS, HbSβ^0^, and HbSD Punjab) and milder genotypes (HbSC and HbSβ).

Serum IgA, IgM, and IgG concentrations were determined every 18–24 months by using a nephelometric technique during systematic complete check-up and entered prospectively into the CHIC database as part of standard patient management. Patient and treatment characteristics such as age, therapeutic intensification (hydroxyurea (HU), chronic transfusion program (TP), or hematopoietic stem cell transplantation (HSCT)), routine blood count, and hemoglobin (Hb), HbF, and HbS levels were also monitored.

### 2.2. Statistical Analysis

Statistical analysis was conducted with STATA v13 (StataCorp, College Station, TX, USA). IgG, IgA, and IgM levels were described with mean, standard deviation, median, minimum, and maximum and compared between groups by Student *t* test (or Wilcoxon–Mann–Whitney test) and parametric one-way ANOVA (or Kruskal–Wallis test). Two-sided *p* < 0.05 was considered statistically significant. Results are given as box-and-whisker plots.

## 3. Results

From 1996 to 2018, 888 children, including 731 with severe SCD genotypes and 157 with milder genotypes, had at least one complete clinical and biological check-up; 86 patients had undergone splenectomy at a median age of 5.3 years (range 3.75–7.9) but 802 had not at the last follow-up. Overall, 4225 IgG, 2875 IgM, and 2876 IgA values were collected and analyzed.

Considering all patients, or restricting the analysis to severe SCD subtypes only, we found no evidence of any effect of Glucose-6-Phosphate dehydrogenase (G6PD) deficiency or α-thalassemia (deletion of one or two α genes) on IgG, IgA, and IgM levels over time.

We next sought to describe the natural history of SCD in terms of Ig profiles and, therefore, restricted our analysis to data collected before any therapeutic intensification and/or splenectomy. The data addressing differences between severe and milder subgroups are shown in Figure 1. All comparisons reached statistical significance, which indicated high IgG level in the severe subgroup except for the youngest population (before 3 years of age, *p* = 0.227). IgA levels were significantly increased for all age groups in the severe SCD subtype. Conversely, we found no consistent differences in IgM levels between both subgroups and noted low IgM levels over time.

The bottom of each box indicates the 25th percentile, the middle line the 50th percentile, and the top of the box the 75th percentile. Vertical lines indicate limits of 1.5 times the interquartile range (*n* indicates the total number of values).

Finally, to evaluate the impact of therapeutic intensification on Ig levels, we limited our analysis to children with severe genotypes and excluded all values measured after splenectomy. Therefore, we compared Ig levels close to therapeutic modification, last and first collected values before and after therapeutic intensification, respectively. We also evaluated Ig levels at last check-up. Only data for children switched from no intensification to the first therapeutic intensification were analyzed. After intensification by TP (279 children), IgG and IgA levels stabilized, whereas IgM levels decreased significantly. After HU introduction (347 children), both IgG and IgA levels increased significantly, and IgM levels decreased significantly (Table 1). Only a few data were available for children switched from no intensification to HSCT, which precluded statistical significance.

To understand the mechanisms contributing to changes in Ig profiles after therapeutic intensification, we sought to analyze Hb, HbF, and HbS levels and white blood cell and neutrophil counts right before and after the first therapeutic intensification. Neutrophil counts were not modified after TP but were significantly decreased after HU introduction. Only data collected at the same time as IgG values are shown in Table 2.

## 4. Discussion

In early childhood (before 3 years of age), we found that IgA levels were significantly increased in the SS/Sb^0^/SD Punjab subgroup, but we observed no different patterns with IgG or IgM. These data are in agreement with those of De Ceulaer, who compared 63 children with HbSS disease from birth to 2 years of age with age-matched control children with HbAA [14]. After 3 years and throughout adolescence, we found consistent sickle genotype differences in IgA and IgG levels, with significantly increased values in the severe subgroup.

Limiting our analysis to data before intensification and/or splenectomy enabled us to explore the natural course of the disease. We confirmed significantly higher IgG and IgA levels over time for the severe genotype subgroup, with a regular and gradual increase to higher than expected levels in the general population. Our findings are consistent with other reports of HbSS disease [11,12,13]. Notably, the largest previously reported study revealed significantly increased IgA and IgG levels in 79 children with HbSS disease who were 1–7 years of age and followed from birth as compared with 114 age/sex-matched HbAA control children. In children with HbSS disease, IgA level was significantly elevated from 2 years of age and IgG from 6 years of age [13].

The basis for the gradual increase in IgG and IgA levels in our cohort is speculative. Polyclonal gammopathies may be caused by any reactive or inflammatory process such as recurrent malaria, and increased IgA levels have been reported in children with malnutrition, conditions prevalent in tropical regions of the world. However, most of the children in our cohort were born in France, and our study included a demographically matched comparison group with milder genotypes (HbSC and HbSβ^+^ thalassemia), exhibiting lower IgG and IgA levels than severe sickle genotypes. In addition to infection, polyclonal gammopathy has been associated with liver or connective tissue diseases [15]. Although liver diseases (autoimmune hepatitis, primary sclerosing cholangitis, and secondary hemochromatosis) and connective tissue diseases can all be observed in patients with HbSS disease, their incidence is very low in children, which precludes any correlation analysis in our cohort.

Elevated levels of IgG and IgA could result from chronic inflammation generated by sickle RBCs through interactions with multiple blood and immune cell populations. There is expanding evidence for important roles of cell types not affected by the sickle mutation, such as leukocytes and endothelial cells, in the SCD pathophysiology [16]. Higher inflammation in HbSS than HbSC disease may account for the different evolution of Ig profiles in these populations. Alternatively, earlier and consistent spleen deterioration in the most severe genotype subgroup could affect the immune functions of the spleen, in addition to altering its filtering function [2].

Considering IgM, we observed a progressive decline in serum IgM levels over time to lower than expected levels in the general population. Low serum IgM level was not more severe in children with than without splenectomy. Conversely, IgM level was not lower in HbSS children than HbAA controls in the prospective Jamaican study and was even higher in a Nigerian study [12,13]. The authors postulated that plasmodium infection led to increased serum IgM level in individuals living in endemic regions. Of note, like in our study, other studies indicating low IgM level in HbSS disease were all performed in nontropical world regions [12]. In these studies, the authors suggested a correlation between loss of splenic tissue and low IgM concentration [10]. Surprisingly, although loss of splenic function occurs later in life and to a lesser extent in HbSC than HbSS [5,6], we found no significant difference in IgM level between the two sickle subtype groups. Moreover, therapeutic intensification (by TP or HU) did not prevent decreased IgM level. Splenomegaly is commonly observed in children with HbSS undergoing TP, notably as a consequence of RBC accumulation. HU has been also associated with splenomegaly among patients with HbSS [17]. Finally, splenomegaly is more likely in patients with HbSC than HbSS [5]. We hypothesize that profound disorganization in the spleen (as shown on histopathologic reviews [2]), not only of the red pulp but also the white pulp containing the B-cell compartment, by congested RBCs, could account for the low IgM level in children with a persistent splenomegaly. In autoimmune lymphoproliferative syndrome (ALPS) where splenomegaly develops early in life because of the accumulation of T-lymphocytes, low serum IgM is frequently noted and children display a B-cell deficiency that is related to disorganization of the splenic MZ [18]. Notably, as in APLS patients, as compared with 10 patients with SC disease or 10 control patients, 8 patients with HbSS disease showed significantly lower circulating MZ B cells (IgM memory B cells, unique to the spleen and critical for the phagocytosis of encapsulated bacteria) [7]. Certainly, spleen tissues from children with SCD should receive more attention, and immunohistochemistry studies using anti-CD20 antibodies, in addition to splenocyte analysis by flow cytometry, should be performed. This search might help in understanding the mechanism by which SCD results in low serum IgM levels. Additionally, circulating B-cell subsets should be further investigated during the course of SCD to increase our knowledge of the pathophysiological mechanisms associated with the increased vulnerability to infections in children with SCD.

In our cohort, therapeutic intensification with HU or TP was introduced at about 5 years of age (mean age 5.4 years for HU and 5.7 years for TP). IgA and IgG levels significantly increased after the introduction of HU but stabilized in children receiving a TP. When we compared Hb levels, neutrophil count, and HbS levels obtained with HU and TP, HbS levels were significantly reduced with TP. Conversely, neutrophil count was significantly reduced with HU (Table 2). Our results suggest that low HbS level versus low neutrophil count may be a better way to reduce inflammation.

Surprisingly, HU therapy failed to prevent increased IgG and IgA level over time. Unfortunately, our study did not address adherence to medication. Another limitation was that the HU dosage was not recorded in our database. HU was usually given at 12 mg/kg per day during the first month and then 23 mg/kg per day but rarely increased until the maximally tolerated dose, as reflected by the moderate myelosuppression observed in our cohort (mean absolute neutrophil count 3.9 ± 1.9 g/L at last check-up). Whether HU dosage escalation to the maximal tolerated dose may prevent progression of IgA and IgG level over time remains to be determined. In 2014, HU became the standard of care for children as soon as 9 months of age according to the US National Institutes of Health after a clinical trial of HU for very young children with SCD [4,19]. Further study will be required to investigate the effects of earlier introduction of HU on Ig level.

## 5. Conclusions

We report the largest prospective longitudinal study of IgG, IgA, and IgM levels in a cohort of children with SCD. Without intensification, IgG and IgA levels were significantly higher in the SS/Sβ^0^/SD Punjab than SC/Sβ^+^ subgroup, which suggests higher inflammation. Children receiving chronic transfusion might experience less inflammation, which could explain the stabilization of IgG and IgA levels in this population. We report low serum IgM levels over time, regardless of sickle genotypes and therapeutic intensification. Additional studies are warranted to better understand the mechanisms contributing to these changes in Ig levels and their clinical significance.

## Figures and Tables

**Figure 1 jcm-08-01688-f001:**
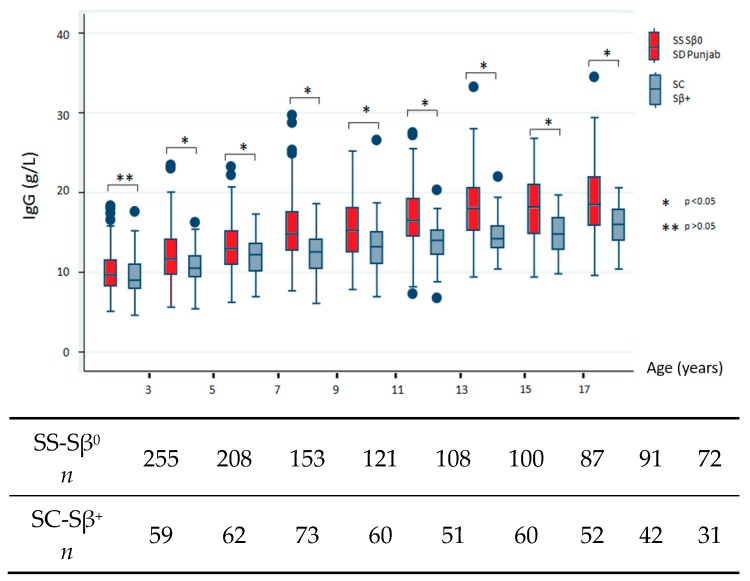
Immunoglobulin (Ig)G, IgA, and IgM levels by age group. Only values collected before any therapeutic intensification and/or splenectomy were analyzed and compared between children with severe sickle genotypes (HbSS, HbSβ^0^, and HbSD Punjab) and milder genotypes (HbSC and HbSβ^+^).

**Table 1 jcm-08-01688-t001:** Comparison of IgG, IgA, and IgM levels right before and after first therapeutic intensification (HbSS, HbSβ^0^, and HbSD Punjab only, and excluding values collected after splenectomy). Only data for children switched from no intensification to the first therapeutic intensification were analyzed.

		IgG		IgA		IgM	
		Age (years)	IgG Level (g/L)	Age (years)	IgA Level (g/L)	Age (years)	IgM Level (g/L)
**Transfusion Therapy**	**Before treatment**	5.6 ± 4.7	12.52 ± 4.71	4.8 ± 4.3	1.65 ± 1.06	4.8 ± 4.3	1.04 ± 0.60
	**1st value after intensification**	7.8 ± 4.8	13.12 ± 4.62	7.5 ± 4.5	1.67 ± 0.90	7.5 ± 4.5	0.75 ± 0.34
	**Latest value**	9.7 ± 4.7	13.70 ± 3.83	9.6 ± 4.4	1.72 ± 0.80	9.6 ± 4.4	0.73 ± 0.45
			*p* = 0.2645		*p* = 0.935		***p* = 0.0017**
**Hydroxyurea**	**Before treatment**	5.4 ± 3.8	12.60 ± 3.80	5.3 ± 4.0	1.60 ± 0.90	5.3 ± 4.0	1.04 ± 0.4
	**1st value after intensification**	8.9 ± 4.3	13.90 ± 4.0	8.8 ± 4.5	2.10 ± 0.80	8.8 ± 4.5	0.85 ± 0.33
	**Latest value**	11.3 ± 4.1	14.90 ± 4.30	10.9 ± 4.0	2.40 ± 1.00	10.9 ± 4.0	0.80 ± 0.37
			***p* = 0.0001**		***p* < 0.0001**		***p* = 0.0017**

Data are mean ± SD. Bold values indicate significance at *p* < 0.05.

**Table 2 jcm-08-01688-t002:** Comparison of hemoglobin (Hb), HbF, and HbS levels and white blood cell (WBC) and polynuclear neutrophil (PNN) counts right before and after the first therapeutic intensification (HbSS, HbSβ^0^, and HbSD Punjab only). Data were collected at the same time as IgG values. Only data for children switched from no intensification to the first therapeutic intensification were analyzed.

		Age (years)	Hb Level (g/dL)	HbF Level (%)	HbS Level (%)	WBC Count (G/L)	PNN Count (G/L)
**Transfusion Therapy**	**Before treatment**	5.6 ± 4.7	7.9 ± 1.1	12.4 ± 7.3	74.4 ± 13.2	13.4 ± 4.8	5.8 ± 2.4
	**1st value after intensification**	7.8 ± 4.8	9.1 ± 1.2	4.9 ± 3.8	38.1 ± 15.6	11.9 ± 3.8	6.1 ± 2.8
	**Latest value**	9.7 ± 4.7	9.1 ± 1.0	4.4 ± 4.9	33.7 ± 11.3	11 ±4.3	5.9 ± 2.3
			***p* < 0.0001**	***p* = 0.0001**	***p* = 0.0001**	***p* = 0.001**	***p* = 0.5791**
**Hydroxyurea**	**Before treatment**	5.4 ± 3.8	8.1 ± 1.2	11.8 ± 7.5	75.1 ±13.4	13.1 ±4.9	6.3 ± 4.5
	**1st value after intensification**	8.9 ± 4.3	8.8 ± 1.2	17.1 ± 8.9	70.38 ± 11.8	8.5 ± 3.9	4.5 ± 4.9
	**Latest value**	11.3 ± 4.1	9.1 ± 1.2	15.8 ± 9.2	68.8 ± 15.7	7.4 ± 3	3.9 ± 1.9
			***p* < 0.0001**	***p* < 0.0001**	***p* = 0.0007**	***p* = 0.0001**	***p* = 0.0001**

Data are mean ± SD. Bold values indicate significance at *p* < 0.05.

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
