# Peer review of "Serum Immunoglobulin Levels in Children with Sickle Cell Disease: A Large Prospective Study"

_jcm, 2019, doi:10.3390/jcm8101688_

Round 1

Reviewer 1 Report

This is an interesting study investigating IgA, IgG and IgM serum levels in pedatric patients with SCD or SCD with Beta thalassemia.  The authors data show high levels of IgA and IgG in the severe compared to milder genotypes, but lower / suppressed IgM levels irrespective of sickle genotype. 

The strengths of the studies are the large cohort base (888 pediatric cases). The primary weaknesses is the english grammar. Qualifying statements such as "the largest prospective longitudinal ......" should be changed to " A large prospective ....."  

Author Response

Please find our revised manuscript “Serum immunoglobulin levels in children with sickle cell disease: a longitudinal prospective study” that we are submitting for publication as a brief report in Journal of Clinical Medicine, for the Special Issue “Sickle Cell Anemia: From Genetic Epidemiology to New Therapeutic Strategies”. Manuscript ID: jcm-592791

We have carefully analyzed the comments of the two reviewers and substantially revised the manuscript according to their comments. Both the introduction and discussion sections have been implemented with the results of prior studies, and we developed further the significance of our findings. Notably, only few and old studies have been published on immunoglobulin levels in sickle cell disease, and the mechanisms contributing to these changes remain to be elucidated.

English changes required have been made, with the help of an English-language editor.

Finally, references of prior literatures have been added.

Reviewer 2 Report

In this work, the authors reported the largest prospective longitudinal study of IgG, IgA and IgM levels in a cohort of children with SCD in different subgroups. The authors find IgG and IgA levels were significantly higher in the SS/Sβ0/SDPunjab subgroup, compared to the SC/Sβ+ subgroup while Low serum IgM levels over 184 time were observed, irrespective of sickle genotypes and therapeutic intensification.  The contents of the manuscript lie within the scope of JCM, but could be accepted if the following comments can be addressed and the respective replies and clarifications have been incorporated into the revised manuscript.

The introduction part is too simple and brief. The authors should provide more information on the physiological functions IgG, IgA, and IgM and how they are varied in SCD in different subgroups in prior studies and how they are connected to the disease severity or clinical manifestations of SCD. Only five papers were mentioned in the entire introduction section and 11 references overall, suggesting prior literatures are not reviewed thoroughly.  In the discussion, the authors should connect the findings with clinical evidence to provide new hypothesis on the pathogenesis or therapeutic interventions to improve the clinical significance of this paper.

Author Response

(The authors gave the same response as above.)

Round 2

Reviewer 2 Report

The authors have addressed all the concerns I have.